# Case Series of Acute Peripheral Neuropathies in Individuals Who Received COVID-19 Vaccination

**DOI:** 10.3390/medicina59030501

**Published:** 2023-03-03

**Authors:** Eglė Sukockienė, Gautier Breville, Damien Fayolle, Umberto Nencha, Marjolaine Uginet, Annemarie Hübers

**Affiliations:** Department of Clinical Neurosciences, Division of Neurology, Geneva University Hospitals, 1205 Geneva, Switzerland

**Keywords:** vaccination, COVID-19, neuropathy, Guillain–Barré syndrome, Parsonage–Turner syndrome

## Abstract

*Background and Objectives*: Vaccination has been critical to managing the COVID-19 pandemic. Autoimmunity of the nervous system, especially among a select set of high-risk groups, can be triggered or enhanced by the contents of vaccines. Here, we report a case series of acute peripheral neuropathies following vaccination against severe acute respiratory syndrome coronavirus 2 (SARS-CoV-2). We report on 11 patients (range: 30–90 years old) who presented at our center between January 2021 and February 2022. *Methods*: We obtained the patients’ history and performed clinical neurological examination and electromyoneurography on all subjects. If necessary, magnetic resonance imaging and laboratory testing, including cerebrospinal fluid analysis and specific antibody testing, were performed. *Results*: Patients presented with peripheral neuropathies of acute onset between 1 and 40 days after vaccination with different types of COVID-19 vaccines. Most cases (9/11) resolved with a rapid, complete or partial recovery. *Conclusions*: We found acute peripheral neuropathies in a set of individuals after they received vaccines against SARS-CoV-2. Albeit our observation shows that during extensive vaccination programs, negative side effects on the peripheral nervous system might occur, most of them showed benign clinical evolution. Thus, potential side effects should not hinder the prescription of vaccines. More extensive studies are needed to elucidate populations at risk of developing peripheral neuropathies and mechanisms of autoimmune response in the nervous system.

## 1. Introduction

Side effects from vaccinations can be associated with neurological disorders, such as acute disseminated encephalomyelitis or brachial neuropathy. Whether there is a causal relationship between different vaccines and other peripheral nervous system disorders, such as Guillain–Barré syndrome (GBS) or Bell’s palsy, remains under debate [1]. While the spectrum of neurological complications following SARS-CoV-2 infection has been well documented [2], little is known regarding potential side effects of the mRNA- and viral-vector-based SARS-CoV-2 vaccines. Common side effects of these vaccines are local (pain at the injection site, local cutaneous symptoms, oral symptoms, local myalgia) but can be systemic, too (rash, headache, generalized myalgia, fever and shivering) [3]. Several studies have shown that local side effects are more frequent in the mRNA-based vaccine group, while systemic side effects are more frequent in the viral-vector-based vaccine group [3,4,5]. In addition, a younger age and the female sex seem to be risk factors for developing generalized side effects [3,4,6,7].

During the worldwide pandemic, recommendations have been proposed by different national neurological societies regarding the indication of SARS-CoV-2 vaccines in patients suffering from neuropathies. Yet, a better understanding of potential side effects of these novel SARS-CoV-2 vaccines on the peripheral nervous system is needed. This is an important issue that needs to be addressed, as many open questions and insecurities continue to arise with respect to SARS-CoV-2 vaccination among the general public, patients, and health care professionals, especially as planned annual vaccine programs are being scheduled.

Here, we report on a series of 11 patients presenting at a single center between January 2021 and February 2022 with acute disorders of the peripheral nervous system after vaccination against SARS-CoV-2 with either BNT162b2, mRNA-1273, or Ad26.COV2-S.

## 2. Materials and Methods

Bioethics approval for a medical chart review was provided by the Commission Cantonale d’Ethique de la Recherche (CCER). Inclusion criteria: patients who presented to the neuromuscular department at the Geneva University Hospitals between 1 January 2021 and 1 February 2022 with an acute condition affecting the peripheral nervous system after vaccination against SARS-CoV-2. Exclusion criteria: patients with an absence of an electroneuromyography (ENMG) recording or refusal to participate in the study. Eleven patients (five female, mean age 66 ± 16 years, median age 67 years) were identified. Charts were assessed for demographics, medical history, diagnosis, vaccine type, laboratory results, ENMG results (signs of demyelination or axonopathy), imaging results (contrast enhancement or enlargement of roots and nerves on magnetic resonance imaging (MRI)), medical management (treatment with intravenous immunoglobulin (IVIG), plasmapheresis, steroids and anti-infectious medications), and response to treatment and clinical outcomes (hospital stay or management in outpatient setting). No statistical analyses were performed but the results are reported in Table 1.

## 3. Results

From 1 January 2021 to 1 February 2022, administered doses of vaccines against SARS-CoV-2 in the canton of Geneva [8] were Pfizer BioNTech (BNT162b2): 371,527; Moderna (mRNA-1273): 561,367; Johnson&Johnson/Janssen Ad26.COV2-S: 2204. The clinical data of the patient cohort are summarized in Table 1. None of the patients was on neurotoxic treatment and none of them had neurotoxic medications in the past. All patients presented with a peripheral neuropathy of acute onset and with a timely relation to SARS-CoV-2 vaccination. Three patients presented with unilateral peripheral facial palsy, two with GBS (one of which with additional facial biplegia), one with acute axonal sensory neuropathy, four with acute brachial neuropathy (Parsonage–Turner syndrome, PTS), and one with phrenic nerve palsy. Unilateral symptoms affecting the upper extremities were, in all cases, located at the injection side of the body. Ten of eleven patients were from Switzerland, one from neighboring France. Eight of eleven patients were from Geneva canton, one from Vaud canton, and one from Bern canton.

The mean (± standard deviation) time to onset was 11 ± 12 days (see Table 1); the median time to onset was 7 days. 

On electrophysiological examination, patients with peripheral facial palsy had signs of demyelination, with little or no axonal loss. Patients with GBS showed: (1) reduced excitability of motor and sensory nerves without signs of acute denervation (i.e., reduced amplitudes of nerve potentials after peripheral electrical stimulation, but no pathological spontaneous activity at needle myography); (2) signs of demyelination of the facial nerve following a conduction study in the case of bilateral facial palsy, and normal ENMG in the limbs. Patients with acute sensory ataxic neuropathy had a reduction in sensory nerve action potentials in all limbs without any nerve conduction velocity slowing and abolition of the soleus Hoffman reflex (H reflex). In the group of patients with acute brachial neuropathy (PTS), ENMG showed axonal loss in various patterns in the brachial plexus distribution: (1) upper trunk, (2) axonal loss in C5–C6 segments and clinical phrenic nerve involvement, (3) C8–Th1 segments, and (4) normal ENMG (13 days after symptom onset).

In general, patients showed a benign evolution with, in the majority of cases, almost complete resolution of their symptoms at follow-up. 

All patients with unilateral peripheral facial palsy received prednisone 0.5–1 mg/kg/d and valacyclovir per os for seven to ten days, following a local protocol. Two patients with PTS received oral prednisone following a dose reduction scheme (starting dose: 1 mg/kg/d) over four weeks. Only one of the patients presenting with GBS was treated with intravenous immunoglobulin (IVIG). 

## 4. Discussion

Here, we report on a series of 11 patients who presented a wide spectrum of acute peripheral inflammatory neuropathies in timely association with SARS-CoV-2 vaccination, including BNT162b2 and mRNA-1273, and one case related to the Ad26.COV2-S vaccine. No cases in relation with ChAdOx1 nCoV-19 vaccination were observed, which is most certainly explained by the fact that this product was not authorized in Switzerland at the time of the study. The association of vaccination with the development of inflammatory neuropathies was based on a close temporal relationship.

There is a long history of reports of some viral vaccines, namely against influenza, associated with, albeit with rarity, cases of GBS (for review, see [1]). With SARS-CoV-2 vaccines, Ad26.COV2-S, a viral vector vaccine, has been associated with GBS in 100 cases in the United States according to the Food and Drug Administration (FDA). Meanwhile, a population-based study of more than 32 million people who received the ChAdOx1 nCoV-19 vaccine found an increased risk of hospital admission for GBS (15–21 days and 22–28 days), Bell’s palsy (15–21 days) and myasthenic disorders (15–21 days) [9]. There have been recent reports concerning a possible association of the BNT162b2 mRNA COVID-19 vaccine with Bell’s palsy [10] and GBS from a large population-based study [11] and a surveillance study [12]. Additionally, an analysis of pharmacovigilance reports found GBS with associated facial palsy following both adenovirus-vectored vaccines and mRNA vaccines [13]. Meanwhile, there have been single case reports of GBS [14] and Bell’s palsy following mRNA-1273 vaccination [15].

Immunization by vaccination activates antigen-specific cellular and/or humoral immunity through a suggested potential mechanism: ‘molecular mimicry’ where epitopes in the vaccine initiate development of antibodies and/or T-cells that cross-react with epitopes on myelin or axonal glycoproteins. Activated macrophages could potentially be targeted to antigens of myelin or nodes of Ranvier [1]. It is possible that host genetic polymorphisms and individual factors have a predisposing role in the development of inflammatory neuropathies as well as in the context of vaccination. For instance, genetic factors involved in the pathogenesis of GBS could be human leukocyte antigen (HLA) genes, cluster of differentiation (CD) 1A, FAS, Fc gamma receptors (FcGR), intercellular adhesion molecule-1 (ICAM1), different interleukins, nucleotide oligomerization domain (NOD), toll-like receptor 4 (TLR4), or tumor necrosis factor-α (TNF-α) [16]. However, the vaccines against SARS-CoV-2 do not contain an exogenous adjuvant. Possibly, the vaccine might induce immune activation from a combined effect of mRNA and lipids in a vaccine, potentially including interferon production, thus triggering an inflammatory response [17].

Only two patients in our group presented with sensory-motor GBS, one very mild case after vaccination with the BNT162b2vaccine, and one case that showed a more severe disease course and required treatment with IVIG. The latter case was in a patient vaccinated with Ad26.COV2-S. One patient presented with a rare case of isolated sensory axonal neuropathy associated with anti-GD1a antibodies. He showed a benign disease course with a complete resolution of symptoms under purely symptomatic pain treatment.

The majority of the patients we report here presented with either peripheral unilateral facial palsy, or PTS. All three patients with facial palsy showed, except for one patient, a very good improvement (i.e., improvement of I to II stages on the House–Brackmann clinical scale) ten days after symptom onset. 

While two out of three patients with facial palsy had received the mRNA-1273 vaccine, all four cases of PTS were in patients who had received the BNT162b2 vaccine. Yet, as our series is small, no conclusion on statistically significant correlations can be drawn. Patients with PTS showed a good improvement of their pain symptoms, but only partial improvement of the associated muscle weakness of the upper limb at the time of follow-up. Of note, PTS always affected the site of the body where the vaccine was injected. Interestingly, one patient presented with orthopnea and dyspnea and was later diagnosed with phrenic nerve palsy. To our knowledge, no previous studies have reported phrenic nerve palsy following vaccine against SARS-CoV-2. 

A retrospective study by Kim et al. found that Bell’s palsy occurred more during the COVID-19 vaccination period than in the three pre-vaccination years. In Bell’s palsy that had a timely relation with COVID-19 vaccination (<42 days, *n* = 13, 12/13 vaccinated with mRNA vaccines), compared to the vaccine-unrelated facial neuropathy, prognosis was better, and the patients were younger [18].

The majority of patients showed quick improvement of their symptoms, requiring only symptomatic treatment. Furthermore, recent results are reassuring in regard to the vaccination recommendations for patients with a history of GBS: none of 162 participants with a previously reported episode of GBS had a recurrence of neuropathy after vaccination with vaccines against SARS-CoV-2, including BNT162b2, ChadOx1/nCoV-19, mRNA-1273, and Ad26.COV2.S. Patients with chronic autoimmune neuropathies such as chronic inflammatory demyelinating polyneuropathy (CIDP) and multifocal motor neuropathy (MMN) reported mild symptom fluctuations, requiring a modification of maintenance treatment [19]. The only two cases that showed no or only minor improvement were the two youngest patients in our series (male, 30 and 53 years). This is in line with previous studies describing stronger side effects in the younger population [4]. Of course, we note that besides the limited number of patients presented here, one additional constraint is the retrospective character of our study.

In conclusion, our study shows that acute inflammatory neuropathies can be observed in a timely relation to SARS-CoV-2 vaccinations with viral-vector-based as well as with mRNA-based products. Yet, the symptoms present as mild to moderate, with a good and rapid improvement in the majority of cases. We think that this is important information that can help clinicians in the decision-making process regarding the vaccination of different patient groups, but also diminish insecurity and fear with regard to these products in the general population, especially as vaccines are being developed for the new variants of SARS-CoV-2. 

## 5. Conclusions

As has been found with other vaccines, vaccines against SARS-CoV-2 may be associated with transient peripheral neuropathies among a set of individuals.

## Figures and Tables

**Table 1 medicina-59-00501-t001:** Profile of Neuropathy Cases.

Number	Age, Sex	Diagnosis	Time between Symptom Onset and Vaccination	Type of Vaccine	Symptoms	Anti Ganglioside Antibodies	Cerebrospinal Fluid	Comorbidities	Imaging	Treatment	Outcome
1.	88, female	Brachial neuropathy (PTS)	<24 h after the second dose.	Pfizer BioNTech (BNT162b2)	Paresis of left biceps, triceps, brachioradialis. Sharp pain, dysesthesia.	N/A	N/A	Diabetes, Hypothyrosis, Atrial fibrillation, Arterial hypertension, Dyslipidemia	MRI of the left brachial plexus: contrast enhancement.	Symptomatic pain treatment	Good. Discrete paresis of the left triceps (4 in MRC scale) after 8 weeks.
2.	65, female	PFP	~30 h after the first dose.	Moderna (mRNA-1273)	PFP on the left, House–Brackmann Grade II.	N/A	N/A	Hypothyrosis, Dyslipidemia	N/A	Valacyclovir, prednisone 1 mg/kg 7 days.	Good. Slight facial asymmetry 10 days after symptom onset.
3.	53, male	PFP	<24 h after first dose	Moderna (mRNA-1273)	PFP on the left, House–Brackmann Grade III–IV.	N/A	N/A	None	N/A	Valacyclovir, prednisone 1 mg/kg 10 days.	No improvement 10 days after symptom onset.
4.	67, male	Acute sensory axonal polyneuropathy	48 h after the first dose	Moderna (mRNA-1273)	Increasing dysesthesia of all four limbs	Anti-GD1a antibodies 60%	Albumin 0.6 g/L	Pulmonary sarcoidosis, Hepatic cirrhosis	No contrast enhancement on brain and spinal cord MRI.	Symptomatic pain treatment	Good. Complete resolution of symptoms 5 months after onset.
5.	71, female	PFP	~40 days after the first dose	Moderna (mRNA-1273)	PFP on the left, House–Brackmann Grade III–IV.	N/A	N/A	Diabetes, Dyslipidemia, Arterial hypertension	N/A	Valacyclovir, prednisone 0.5 mg/kg 7 days.	Good. House–Brackmann Grade II 10 days after symptom onset.
6.	72, male	Brachial neuropathy (PTS)	~14 days after the first dose	Pfizer BioNTech (BNT162b2)	Paresis of left biceps, triceps, brachioradialis, left phrenic nerve palsy, neuropathic pain.	N/A	N/A	Diabetes, Chronic renal insufficiency, Aortic aneurysm, Hypercholesterolemia, Arterial hypertension	No contrast enhancement on spinal cord MRI.	Prednisone in decreasing dosage, starting from 100 mg/d.	Good. Complete resolution of paresis and pain, slow improvement of breathing difficulties 2.5 months after symptom onset.
7.	30, male	GBS with facial biplegia	~15–20 days after the first dose	Johnson&Johnson/ Janssen Ad26.COV2-S (recombinant)	Facial biplegia	negative	Albumin 1.9 g/L	None	Brain and spinal cord MRI: contrast enhancement on both facial nerves and sacral roots	IVIG 0.4 g/kg for 5 days and cefriaxone	Good. House–Brackmann Grade IV on the right and II on the left after 8 days after symptom onset.
8.	69, male	Brachial neuropathy (PTS)	7 days	Pfizer BioNTech (BNT162b2)	Neuropathic pain on the left upper extremity, paresis of intrinsic hand muscles and wrist extension	N/A	N/A	Cervical disc arthrosis, Lumbar hernia	No contrast enhancement on spinal cord MRI, bulging discs with root compressions C5-C7 bilaterally	Prednisone in decreasing dosage, starting from 1 mg/kg.	Pain completely resolved, paresis persisting 20 days after onset
9.	90, female	GBS	3 weeks after first dose	Pfizer BioNTech (BNT162b2)	Ascending weakness and sensory symptoms in the lower limbs	N/A	N/A	Anemia, Interstitial pneumopathy, Arterial hypertension	N/A	Spontaneous recovery without treatment	Good. Improvement of paresis and walking difficulties
10.	63, female	Brachial neuropathy (PTS)	13 days after the first dose	Pfizer BioNTech (BNT162b2)	Neuropathic pain, hypoesthesia and transient weakness of the left shoulder and arm	N/A	N/A	Hypothyrosis, Chronic obstructive lung disease	Brain, spinal cord and left brachial plexus MRI normal	Symptomatic pain treatment	Lost on follow up
11.	57, male	Phrenic nerve palsy	3 days after the second dose	Moderna (mRNA-1273)	Orthopnea and dyspnea during effort.	N/A	N/A	None	N/A	No treatment	Partial spontaneous improvement of symptoms after 3 months

Abbreviations: MRC—Medical Research Council, PFP—peripheral facial palsy, GBS—Guillain–Barré syndrome, IVIG—intravenous immunoglobulin, PTS—Parsonage–Turner syndrome.

## Data Availability

The data presented in this study are available within the article.

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
