# Peer review of "Case Series of Acute Peripheral Neuropathies in Individuals Who Received COVID-19 Vaccination"

_medicina, 2023, doi:10.3390/medicina59030501_

Round 1

Reviewer 1 Report

Authors reported on a cohort of patients who developed acute peripheral neuropathies, after they received different types of vaccines against SARS-CoV-2.The paper is well written and deserve interest. I have only a few suggestions:

It would be useful to add some comments on the potential pathogenetic mechanisms that could explain the adverse event

Any comorbidities, including pre-existing autoimmune diseases, should be reported in the table

Author Response

It would be useful to add some comments on the potential pathogenetic mechanisms that could explain the adverse event

Thank you for the suggestion. A description of possible pathogenetic mechanism has been added in the discussion:  

Immunization by vaccination activates antigen-specific cellular and/or humoral immunity through a suggested potential mechanism:  ‘molecular mimicry’ where epitopes in the vaccine initiate development of antibodies and/or T cells that cross-react with epitopes on myelin or axonal glycoproteins. Activated macrophages could potentially be targeted to antigens of myelin or nodes of Ranvier (Haber et al., 2009). It is possible that host genetic polymorphisms and individual factors have a predisposing role in a development of inflammatory neuropathies as well as in a context of vaccination. For instance, genetic factors involved in the pathogenesis of GBS could be Human leukocyte antigens (HLA) genes, Cluster of Differentiation (CD) 1A, FAS, Fc gamma receptors (FcGR), Intercellular adhesion molecule-1 (ICAM1), different interleukins, Nucleotide oligomerization domain (NOD), Toll-like receptor 4 (TLR4), or Tumor necrosis factor-α (TNF-α) (Khanmohammadi S et al., 2021). However, the vaccines against SARS-CoV-2 vaccines do not contain an exogenous adjuvant. Possibly, the vaccine might induce immune activation from a combined effect of mRNA and lipids in a vaccine, potentially, including interferon production thus triggering an inflammatory response (Ozonoff et al., 2021).

Any comorbidities, including pre-existing autoimmune diseases, should be reported in the table

Thank you for this useful suggestion. In Table 1, the “Diabetes” column has been replaced with a “Comorbidities” column to clarify this point.

Reviewer 2 Report

Material and methods

1.     Could the authors provide the number of vaccinations at the same period in the region where the patients came from?

2.     From where came the patients? All over Switzerland?

3.     Provide inclusion criteria.

4.     Provide exclusion criteria.

5.     What was assessed from the chart of the patients?

6.     How were the cases statistically analyzed?

Results

1.     Did the patients where use any other medication that could cause neuropathy?

2.     Were all the patients previously healthy?

Others

1.     How did the authors correlate the association between neuropathy and vaccination?

The Reviewer would like to ask the authors what this report brings new to the literature. Why should this report be published?
